# Investigation of reward learning and feedback sensitivity in non-clinical participants with a history of early life stress

Matthew Paul Wilkinson[ORCID], Chloe Louise Slaney, Jack Robert Mellor, Emma Susan Jane Robinson*

School of Physiology, Pharmacology & Neuroscience, University of Bristol, Bristol, United Kingdom

* Emma.S.J.Robinson@bristol.ac.uk

**Data Availability Statement:** All data files are available on the open science framework platform (10.17605/OSF.IO/63E8J).

## Abstract

Early life stress (ELS) is an important risk factor for the development of depression. Impairments in reward learning and feedback sensitivity are suggested to be an intermediate phenotype in depression aetiology therefore we hypothesised that healthy adults with a history of ELS would exhibit reward processing deficits independent of any current depressive symptoms. We recruited 64 adults with high levels of ELS and no diagnosis of a current mental health disorder and 65 controls. Participants completed the probabilistic reversal learning task and probabilistic reward task followed by depression, anhedonia, social status, and stress scales. Participants with high levels of ELS showed decreased positive feedback sensitivity in the probabilistic reversal learning task compared to controls. High ELS participants also trended towards possessing a decreased model-free learning rate. This was coupled with a decreased learning ability in the acquisition phase of block 1 following the practice session. Neither group showed a reward induced response bias in the probabilistic reward task however high ELS participants exhibited decreased stimuli discrimination. Overall, these data suggest that healthy participants without a current mental health diagnosis but with high levels of ELS show deficits in positive feedback sensitivity and reward learning in the probabilistic reversal learning task that are distinct from depressed patients. These deficits may be relevant to increased depression vulnerability.

## 1. Introduction

Early life stress (ELS) is a major risk factor in the development of depression [1–4]. ELS has also been found to lower the threshold of stress required to precipitate depression [5], a key major trigger in healthy populations [6]. Elevated levels of childhood stress lead to widespread functional and morphological alterations in the adult brain with the hippocampus, amygdala and prefrontal cortex being most impacted [7,8]. Amongst other functions, these regions are vital mediators of reward learning: the ability of reward in the environment to modulate future behaviour [9–16]. However, how ELS influences the developing brain to predispose individuals to psychiatric illness is not yet understood [7,17,18].

**Funding:** This work was primarily funded by the BBSRC SWBio DTP PhD programme (www.bbsrc.ukri.org, grant numbers: BB/J014400/1 and BB/M009122/1) awarded to MPW. Additional support was also provided from the Wellcome Trust Neural Dynamics PhD studentship (www.wellcome.org, grant number: 108899/B/15/Z) awarded to CLS. The funders had no role in study design, data collection and analysis, decision to publish, or preparation of the manuscript.

**Competing interests:** The authors have declared that no competing interests exist.

Reward learning deficits have been proposed to be an intermediate phenotype in the aetiology and maintenance of depression [19–22]. While many tasks have been used to probe aspects of reward learning in ELS, few of these have been used in depressed patientso [23]. Two tasks that have been used in depressed patients are the probabilistic reward task, which measures reward learning (PRT [24]) and the probabilistic reversal learning task, which measures reward learning alongside both feedback sensitivity and cognitive flexibility (PRLT [25]). Depressed patients have a reduced bias towards the more highly rewarded stimulus in the PRT, a test of reward learning where participants respond to ambiguous stimuli that are unequally rewarded [21]. Deficits in reward learning measured in the PRT have been observed to both predict the risk of disease development [26] and persistence [20,27] therefore demonstrating a link between depression vulnerability and performance in this task. Another widely used task is the PRLT where participants are presented two stimuli that are probabilistically and unequally rewarded. Following learning of the underlying rule the contingencies switch and depressed patients show impaired accuracy and increased sensitivity to probabilistic negative feedback following the rule change [28,29]. The PRLT can therefore be used to measure parameters relating to reward learning, feedback sensitivity and cognitive flexibility [30,31]. Interestingly in the context of ELS, acute stress has also been observed to impair reward learning in the PRT [32,33] suggesting a potential link between stress, reward processing deficits and depression aetiology. Results from these tasks therefore suggest that reward learning impairments could be a relevant intermediate phenotype in the development of depression and that ELS could have a contributary role.

Previous studies have therefore investigated reward processing deficits in people who have experienced ELS. Hanson and colleagues [34] recruited adolescents with a history of physical abuse who then completed a probabilistic learning task where they showed lower associative learning compared to controls. Changes in reward learning have also been reported within another probabilistic reward task, the probabilistic stimulus selection task (PSST). Women with a history of childhood sexual abuse and a diagnosis of Major Depressive disorder (MDD) showed decreased performance on trials requiring learning of previously rewarded information compared to MDD only and control groups [35]. While there is relatively little literature regarding reward learning in ELS there is much more known about the effects of ELS upon cognitive flexibility and other aspects of reward processing [23,36,37]. Neuroimaging studies have shown that participants with ELS show reduced striatal reward anticipation [38] while results from other behavioural tasks have shown that ELS is associated with reduced reward responsiveness [39] and differences in exploitation strategy [37,40]. Finally, of relevance to the PRLT, studies have observed reduced cognitive flexibility in children that have suffered ELS [36,41] although in tasks other than the PRLT.

Although these studies provide valuable insights, they use different tasks to those previously used to study depressed populations making direct comparisons difficult. Additionally, utilisation of tasks such as the PRT and PRLT, that have been successfully translated for rodent use [16,31,42], means that any insights from these tasks in humans can be used translationally to probe the mechanistic underpinnings. Finally, many of these studies have been carried out in adolescents where the brain is still developing as opposed to mature adults. Studies are therefore needed in adults without a current mental health diagnosis to understand if any reward processing changes are present in individuals at higher risk of mental health disorders.

In this study it was aimed to investigate the relationship between ELS and reward processing using translational tasks that have previously been used in both depressed populations and rodents allowing greater comparability of results. We hypothesised that ELS is associated with impairments in reward processing and feedback sensitivity in an otherwise healthy adult population. Due to the exploratory nature of this study we were not able to make more specific

hypotheses. In order to address this hypothesis we split our investigation into three questions: Firstly, does ELS effect reward learning parameters, secondly, are these changes due to depression symptomology and finally, does ELS interact with either acute or lifetime stress to influence reward learning parameters? Two groups of adult participants that self-reported no current diagnosis of a mental health condition or Parkinson's disease were recruited online and completed a survey of adverse childhood experiences [43] before being split into high and no ELS groups based upon this. Participants completed both the PRT (points based) and PRLT to enable comparisons between the different elements of reward learning measured in the tasks. The PRLT data was additionally analysed using a Q-learning model to probe reward learning parameter changes. Participants were asked about stress exposure to enable exploratory analysis investigating if life stress interacts with ELS to cause reward processing deficits. Participants with a history of ELS showed evidence for decreased positive feedback sensitivity in the PRLT, however neither high ELS nor control participants showed a response bias in the points based PRT. As this was the first reported attempt to utilise the PRT in an online environment we recruited an additional population of control participants using a modified task design using direct compensation to validate that the PRT can be successfully performed online.

## 2. Methods

All procedures were approved by the Faculty of Life Sciences and Faculty of Science Research Ethics Committee at the University of Bristol and were performed in accordance with the declaration of Helsinki in addition to all other institutional and national guidelines. The study protocol was pre-registered (www.osf.io/538yk). All participants provided full written consent for the collection, analysis and publication of their data which is available open access (www.ofs.io/63e8j) and were reimbursed at a rate of £6.00 per hour.

### 2.1 Participants

In order to recruit the final study population a total of 586 participants were recruited using the Prolific (www.prolific.co) online platform to complete an online screening questionnaire (see S1 Fig for study overview and Table 1 for participant demographics). These participants were 25–65 years of age, fluent in English, resident in the UK and had no mild cognitive impairments or dementia. Participants completed the early life stress questionnaire [43] (ELSQ) which asks if participants had prior exposure to specific adverse childhood experiences (ACEs). Participants were also asked "Have you got a current diagnosis of a mental health disorder or Parkinson's disease?".

Participants who met the inclusion criteria for high ELS or no ELS and did not report a diagnosis of a mental health disorder or Parkinson's were then invited to take part in the main experiment within a week of screening and were allocated into two groups. A no ELS group (n = 65) contained people reporting no ACEs on the ELSQ while a high ELS group (n = 64) consisted of those who reported ≥3 ACEs (estimated to be the top tercile of the population [43]). In this second phase of the experiment participants entered demographic information before completing the MacArthur Scale of Subjective Social Status [44], Beck's depression inventory II [45] (BDI-II), the Snaith Hamilton pleasure scale [46] (SHAPS) and the Holmes and Rahe stress scale [47]. The SHAPS was additionally scored using the SHAPS-C criteria [48]. For the stress scale participants were asked if each event occurred in either their adult life or the last year to provide a measure of both total adult lifetime stress and recent stress. For all stages of the experiment participants were instructed to use a desktop or laptop only and that they should be in a quiet place with minimal distractions. Sample size was estimated for a

**Table 1. Demographic and self-report measures in the study population.**

| Measure | No ELS (n = 65) | High ELS (n = 64) | Test statistic | p |
|---|---|---|---|---|
| Gender (% male) | 44.6 | 37.5 | $\chi^2(2) = 2.5$ | 0.28 |
| Age (years) | 37.3 ± 1.30 | 38.0 ± 1.24 | U = 1936.0 | 0.50 |
| Education (% graduates) | 64.6 | 65.6 | $\chi^2(5) = 4.9$ | 0.43 |
| Ethnicity (% white) | 95.4 | 82.8 | $\chi^2(4) = 8.7$ | 0.070 |
| Relationship status (% single) | 18.5 | 28.1 | $\chi^2(3) = 1.9$ | 0.60 |
| Employment status (% full time) | 64.6 | 60.9 | $\chi^2(5) = 3.5$ | 0.61 |
| Monetary concerns (% agree/strongly agree) | 36.9 | 56.3 | $\chi^2(3) = 4.4$ | 0.22 |
| ELSQ | 0 ± 0 | 4.36 ± 0.17 | - | - |
| Social status | 6.2 ± 0.17 | 5.2 ± 0.21 | U = 1397.5 | **0.001** |
| BDI-II | 9.4 ± 1.0 | 15.2 ± 1.22 | U = 1315.5 | **0.0003** |
| SHAPS | 1.4 ± 0.25 | 2.56 ± 0.32 | U = 1496.5 | **0.004** |
| SHAPS-C | 24.3 ± 0.67 | 26.4 ± 0.86 | $t_{119.4} = -1.92$ | 0.057 |
| Lifetime stress | 472.8 ± 22.4 | 529.2 ± 23.9 | $t_{127} = -1.72$ | 0.088 |
| Last year stress | 111.4 ± 12.3 | 139.8 ± 17.0 | U = 1939.5 | 0.51 |

Values are shown for each group as mean ± standard error with significant p values (p ≤ 0.05) indicated in bold.

medium effect size (Cohen's $d$ = 0.5) and 80% power for a t-test at 64 participants per group. While other studies have investigated different dimensions of ELS (i.e. emotional abuse vs psychosocial neglect) with regards to cognitive outcome [8] the present study was not powered to enable this.

Early life stress was highly prevalent in the study population with only 21.0% of participants having no adverse childhood experiences (ACEs) and 44.4% of the population suffering three or more ACEs in their childhood (see S2 Fig). 16.0% of respondents self-reported a diagnosis of a mental health disorder or Parkinson's with this being associated with a higher ELSQ score (Mann-Whitney, U = 15725, p < 0.0001).

The two study groups were well matched with respect to gender, age, education, ethnicity, relationship status, employment status and the presence of monetary worries (see Table 1). However, high ELS participants had a self-reported lower social status coupled with higher depression scores in the BDI-II and elevated anhedonia scores in the SHAPS questionnaires. There was no evidence of a difference between groups when participants were asked about stress they encountered in both the last year and their adult lives. When the BDI-II scores were classified into either minimal, mild, moderate or severe depression (see S3 Fig [45]) participants from the high ELS group were more likely to be in greater severity depression groupings (chi$^2$, $\chi^2(3)$ = 12.9, p = 0.005). Similarly when SHAPS scores were classified into either normal (≤2) or abnormal (≥3) hedonic responses [46] members of the high ELS group were more likely to have abnormal scores (see S3 Fig, chi$^2$, $\chi^2(1)$ = 6.3, p = 0.012).

## 2.2 Behavioural testing

Following completion of self-report measures, participants completed the Probabilistic reversal learning task [30,49] followed by the Probabilistic reward task [24]. To complete the tasks participants were required to download and install the Millisecond Inquisit web player (Millisecond, US) which ran both tasks using Millisecond Inquisit v6.2.1. Participants were instructed they were able to earn an additional £2.00 for high performance on the behavioural tasks.

**2.2.1 Probabilistic reversal learning task.** The PRLT was conducted as previously described [30,49] using the task from the Millisecond test library [50]. Participants were instructed to choose between a "lucky" (rich) and "unlucky" (lean) pattern to maximise points. Selection of the rich stimulus enabled participants to gain a point 80% of the time and lose a point 20% of the time with the lean stimulus having opposite contingencies. If no stimulus was chosen within 2s then this was classified as incorrect and participants lost a point. Participants progressed in the task until meeting the reversal criterion, following this the contingencies reversed such that the rich stimuli became lean and vice versa. This criterion was set randomly between 10 to 15 consecutive correct rich choices to stop participants counting to the criterion. Participants first completed a practice phase where they had to achieve the criterion for a single reversal before proceeding to the main task which was completed in three blocks each limited to 9 minutes where participants could reverse as many times as able within a block. Participants who did not pass the practice phase were excluded from analysis.

Data were analysed as previously described [51]. Win-stay probability was defined as the probability that if a participant was rewarded for selecting a stimulus they would select the same stimulus for the next trial. Lose-shift probability was conversely the probability that if a participant lost a point at a stimulus they would switch to the opposite stimulus for the next trial. This enabled win-stay and lose-shift probabilities to be used as measures of positive and negative feedback sensitivity respectively. These were subdivided into either true feedback which matches with the underlying task rules (e.g. being rewarded for selecting the rich stimulus), or misleading feedback (e.g. being rewarded for selecting the lean stimulus) which opposes the underlying task rule. The number of rule changes, how many times participants were able to meet criterion for a rule change, accuracy and response latency per block were additionally analysed. A Qlearn reinforcement learning model was applied to data as previously described [51,52]. The model enables assessment of the ability of participants to integrate feedback over time (learning rate parameter) and the degree of choice variability that participants have when selecting stimuli (beta, low values represent more random choices while high values show a deterministic choice strategy). Additionally it is possible to use the model to generate a perfect strategy given the same inputs as the participants and then compare participant performance to this as a measure of overall reward learning ability to maximise reward (subjective accuracy). Additionally, data per phase (practice, acquisition of the first rule in block 1 and the following two reversals) was analysed consisting of participant accuracy, errors to criterion and win-stay/lose-shift probability.

**2.2.2 Probabilistic reward task.** The PRT was conducted as previously described [24] using the task from the Millisecond test library [53]. Participants were instructed to identify whether the mouth of a presented cartoon face was long or short (≈11% difference in mouth length) to win points over 3 blocks of 100 trials. Participants were shown a face before a mouth was rapidly presented for 100ms with participants given up to 1750ms to respond. Feedback was not provided on every trial but unknown to participants one mouth was rewarded with points three times more often than the other (rich = 60%, lean = 20%). Response key and rich/lean stimuli assignments were counterbalanced across participants and responses that were quicker than 150ms or slower than 1750ms were excluded from analysis. Additional responses that differed by more than 3 standard deviations from the mean following natural log transformation of latencies for each participant were excluded from analysis. Response bias (logB), a measure of reward learning, and discriminability (logD), a measure of task difficulty, were calculated as:

$$logB = \frac{Rich_{correct} \times Lean_{incorrect}}{Rich_{incorrect} \times Lean_{correct}} \qquad (1)$$

$$lodD = \frac{Rich_{correct} \times Lean^{correct}}{Rich_{incorrect} \times Lean_{incorrect}} \qquad (2)$$

**2.2.3 Directly rewarded probabilistic reward task.** Because neither group in the points rewarded PRT showed a response bias we recruited an additional cohort of 81 participants to test the hypothesis that if the reward modality was restored back to the direct monetary reward as used in laboratory tasks the PRT could be successfully carried out online. Details of this additional study are available in S1 File.

## 2.3 Data analysis

Demographic and self-report measures were compared between groups using either $\chi^2$, t-tests or Mann-Whitney U tests where appropriate. Analysis of the main study was structured into three layers starting with a primary analysis for each measure directly comparing between no ELS and high ELS groups. Where data were not normally distributed then efforts were first made to transform data to normality and where this was not possible Mann-Whitney U tests were completed. Normality was assessed by visual inspection of the QQ-plot alongside use of the Shapiro-Wilk test. Win-stay by block data were transformed using the bestNormalize package in R [54]. Where measures were split by a within subject factor such as block or feedback type these were analysed with repeated measures ANOVAs. Where Mauchly's test identified a violation of the Sphericity assumption then this was corrected using the Huynh-Feldt correction. Two sample unpaired t-tests were used for direct group comparisons.

The second stage of analysis aimed to assess the effect of ELS upon a parameter while controlling for the effects of depression symptomology. Due to differences in social status, BDI-II score and SHAPS score between the no ELS and high ELS groups, principal component analysis (PCA) was conducted to reduce the dimensionality of these variables (see S1 and S2 Tables). Because only principal component 1 (PC1) differed between groups and explained 94.6% of variance this was used in ANCOVAs (analysis of covariance) to analyse whether parameter changes were due to ELS or due to changes in depression symptomology accounted for by the PC1 component.

Finally, an exploratory analysis stage was used to understand if stress and gender interacted with ELS to modify reward learning. For this stage generalised linear mixed models (GLMMs) were used due to the number of factors in the analysis. GLMM models were built containing the factors: gender, ELS, lifetime stress, last year stress and age. Models were fit using the glmmTMB package in R 4.0 [55,56] with model refinement conducted utilising stepwise deletion based upon Akaike information criterion before being compared with a null model to protect against overfitting. PC1 was also added to each model following final model selection to assess the effects of depression symptomology. To ensure that any differences between primary/secondary and exploratory analyses were not due to differences in statistical approach, analysis was also repeated using GLMMs with the results being consistent between the two approaches (see S5 Table).

Statistical analysis was conducted in SPSS v26 (IBM, US), MATLAB 2018a (Mathworks, USA) and R 4.0 [55] with output graphics constructed in GraphPad Prism 8 (GraphPad, US). For all analysis $\alpha$ was set at 0.05. All data is shown as mean ± SE with a bar and stars showing evidence of a main effect of ELS in the primary analysis. $^* \leq 0.05$, $^{**} < 0.01$, $^{***} < 0.001$, $^{****} < 0.0001$.

## 3. Results

### 3.1 Probabilistic reversal learning task

In the overall reward learning measures of the number of rule changes that participants were able to complete (Fig 1A) and accuracy (Fig 1B) there was no evidence for a difference between no and high ELS groups. However, participants with a history of high ELS did have a slower average response latency (Fig 1C, RM-ANOVA, $F_{1,126} = 5.03$, $p = 0.027$) with both groups getting equally faster over the course of the three blocks (RM-ANOVA, $F_{1.88,236.7} = 16.1$, $p < 0.0001$). In secondary analysis, designed to probe if parameter changes were due to ELS or underlying depression symptomology, there was no evidence of an effect of depression symptomology (RM-ANCOVA, PCA1: $p > 0.05$) with the main effect of ELS persisting (RM-ANCOVA, ELS: $F_{1,125} = 4.9$, $p = 0.028$). Exploratory analysis intended to probe if a parameter change was mediated by an interaction between ELS and stress measures was conducted on overall reaction times. This did not replicate a main effect of group but did observe older participants having slower reaction times (GLMM, $Z = 2.8$, $p = 0.005$). This analysis also indicated weak evidence of an interaction between group and lifetime stress (GLMM, $Z = 1.55$, $p = 0.065$) but further investigation did not reveal an effect of lifetime stress in either group.

When data were analysed using the Q-learning reinforcement learning model a trend emerged towards high ELS participants having a lower learning rate compared to the no ELS study population (Fig 1D, t-test, $t_{127} = 1.78$, $p = 0.077$). Secondary analysis revealed no effect of PCA component 1 upon learning rate but removed any evidence for an effect of ELS. In exploratory analysis a main effect of ELS was observed (GLMM, $Z = 2.1$, $p = 0.037$) with the addition of PC1 impairing model fit ($\Delta$AIC = 1.69, $\chi^2(1) = 0.31$, $p = 0.57$). Additionally, a relationship between stress in the last year and learning rate was observed whereby increased stress in the last year decreased learning rate (GLMM, $Z = -2.3$, $p = 0.024$). There was no difference in choice variability (Fig 1E) or accuracy compared to a model predicted perfect strategy (Fig 1F) between groups.

Participants with a history of high ELS exhibited reduced positive feedback sensitivity (PFS, Fig 2A, RM-ANOVA, $F_{1,122} = 10.4$, $p = 0.002$) which persisted once depression symptomology was accounted for using PCA component 1 (RM-ANOVA, $F_{1,121} = 6.6$, $p = 0.01$). Exploratory analysis revealed an interaction between ELS and both lifetime stress (GLMM, $Z = -2.15$, $p = 0.031$) and last year stress (GLMM, $Z = -1.99$, $p = 0.047$). Further investigation revealed effects of both stress types upon PFS in the low ELS group only (GLMM, lifetime stress: $Z = -2.35$, $p = 0.019$, last year stress: $Z = -2.2$, $p = 0.026$). In this low ELS group higher lifetime stress was associated with greater PFS but higher stress in the last year was associated with decreased PFS. However it should be noted that although all suggested terms were removed from the model the overall model was a poorer fit than the null when measured by AIC ($\Delta$ AIC = 7.3, $\chi^2(13) = 18.7$, $p = 0.13$).

The effect of ELS upon PFS was consistent across feedback that matched (true feedback) or clashed (misleading feedback) with the underlying task rules (Fig 2B, Mann-Whitney U, true: U = 1443, $p = 0.03$; misleading: U = 1337, $p = 0.005$). This effect appeared to be constrained to PFS with no corresponding changes in lose-shift probability between no ELS and high ELS groups (Fig 2C and 2D).

When initial learning in the PRLT task was assessed, it was apparent that although ELS and control participants performed similarly during the practice phase there was a learning deficit during acquisition of the first reversal criterion in block 1 as evidenced by increased errors to criterion (Fig 3A, Mann-Whitney U, U = 1580, $p = 0.045$) and decreased accuracy (Fig 3B, Mann-Whitney U, U = 1584, $p = 0.036$). Both groups of participants however performed equally well at achieving criterion for a second and third reversal. Unlike the overall measures

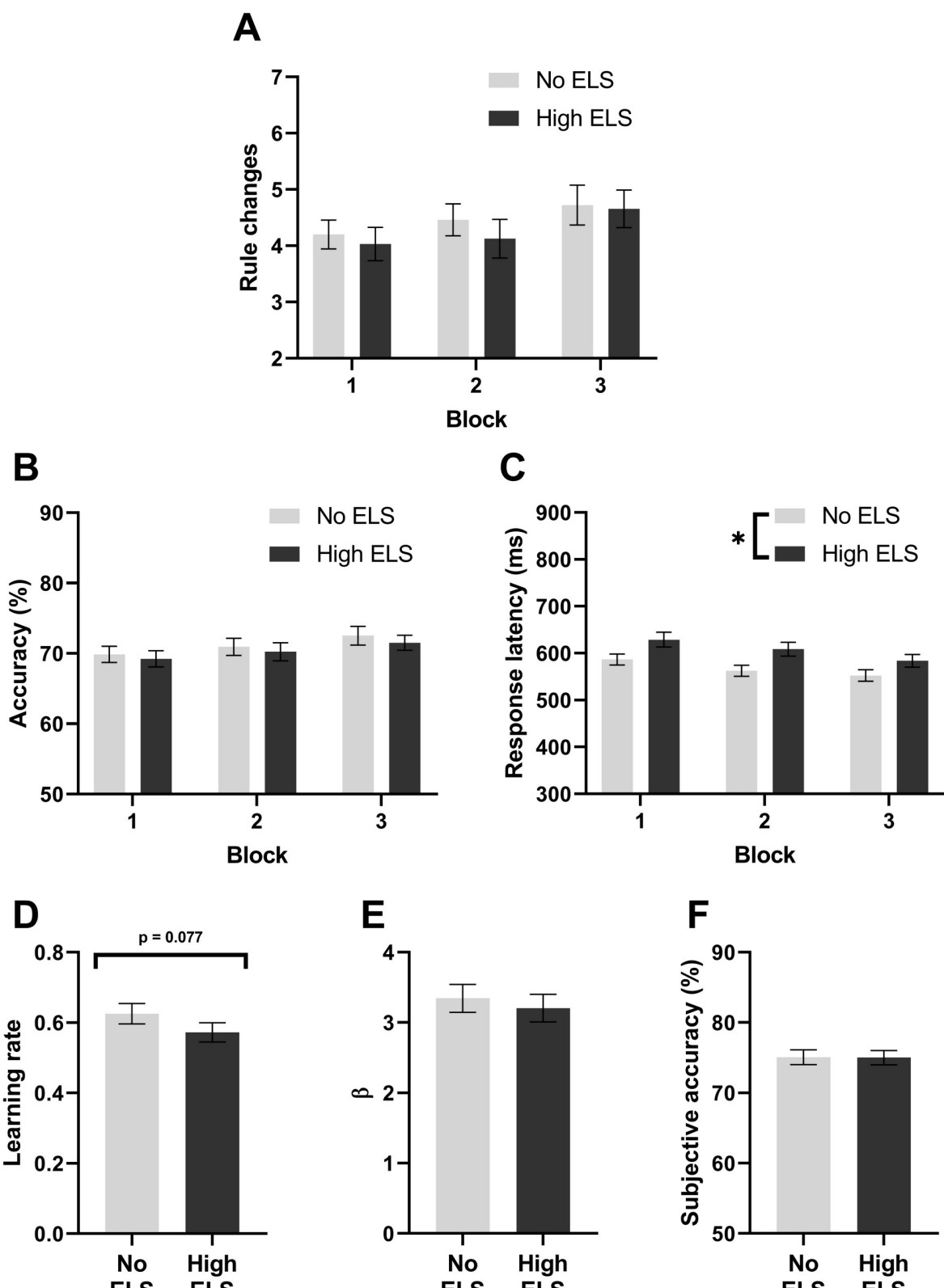

**Fig 1. Overall reward learning and reinforcement learning in the PRLT.** (**A**) Rule changes within each block, (**B**) accuracy by block and (**C**) average response latency per block. From the Q-learning reinforcement learning model: (**D**) learning rate, (**E**) β, the inverse of the softmax temperature and a measure of choice variability and (**F**) subjective accuracy, participant accuracy compared to a model predicted perfect strategy.

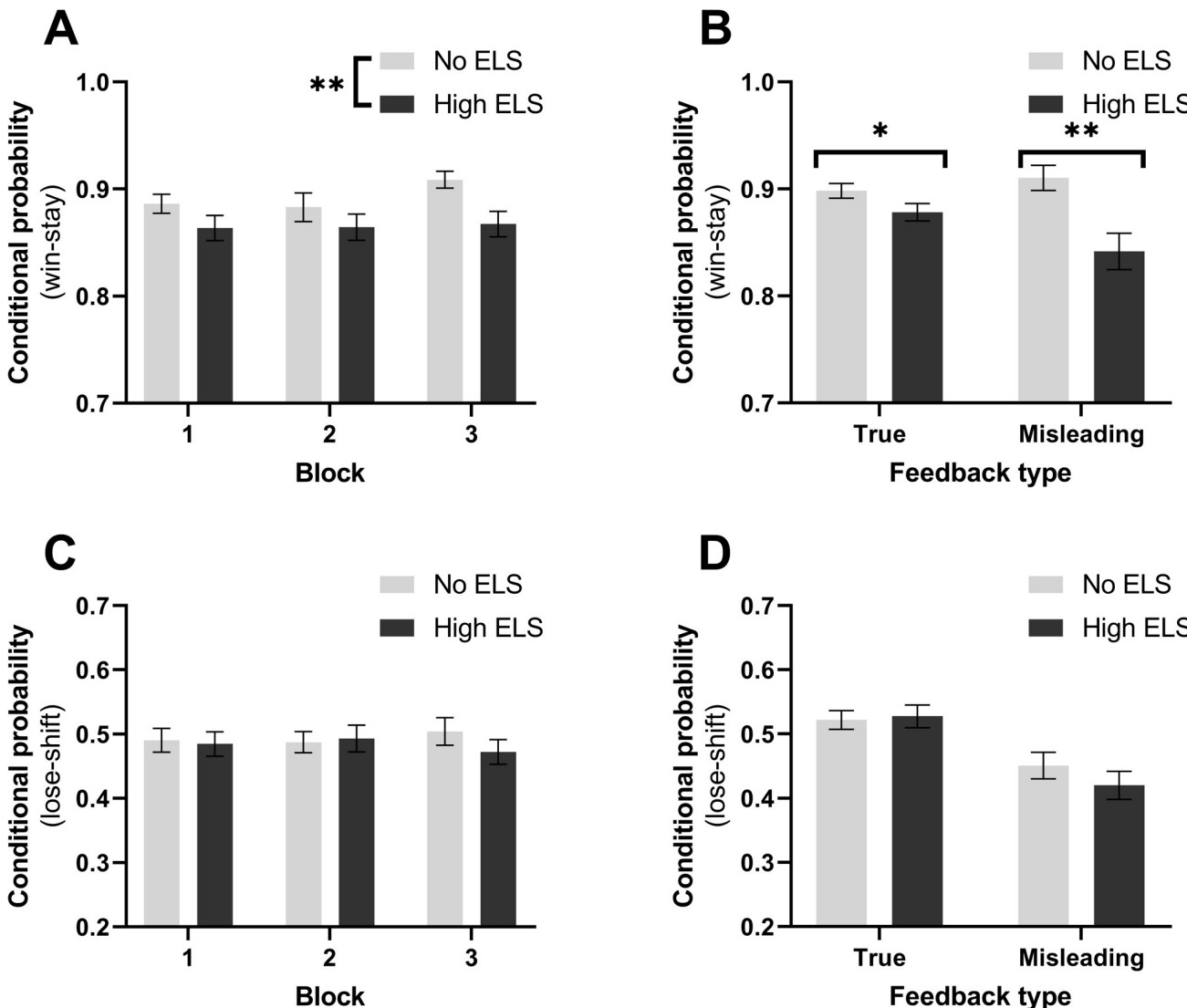

**Fig 2. High ELS participants exhibited lower positive feedback sensitivity than those without a history of ELS.** Win-stay probability overall **(A)** and subdivided into true and misleading feedback **(B)**. Overall Lose-shift probability **(C)** and additionally subdivided into true and misleading feedback **(D)**.

there was no difference in win-stay probability between groups (Fig 3C), however there was a trend for high ELS participants to show increased negative feedback sensitivity (NFS) in the practice phase (Fig 3D, Mann-Whitney U, U = 1532, p = 0.052).

## 3.2 Probabilistic reward task

There was no evidence that participants developed a response bias towards the more highly rewarded stimulus in any block (Fig 4A) nor was there evidence for a response bias developing between blocks (Fig 4B). However, participants with a history of high ELS did show an impaired ability to discriminate between stimuli (Fig 4C, ANOVA, $F_{1,127} = 4.8$, p = 0.030). Secondary analysis revealed that this difference between groups appeared to be driven by differences in depression symptomology with the effect of ELS disappearing when PCA component 1 was included in the analysis (ANCOVA, PCA1: $F_{1,126} = 6.08$, p = 0.015; ELS: $F_{1,126} = 1.7$,

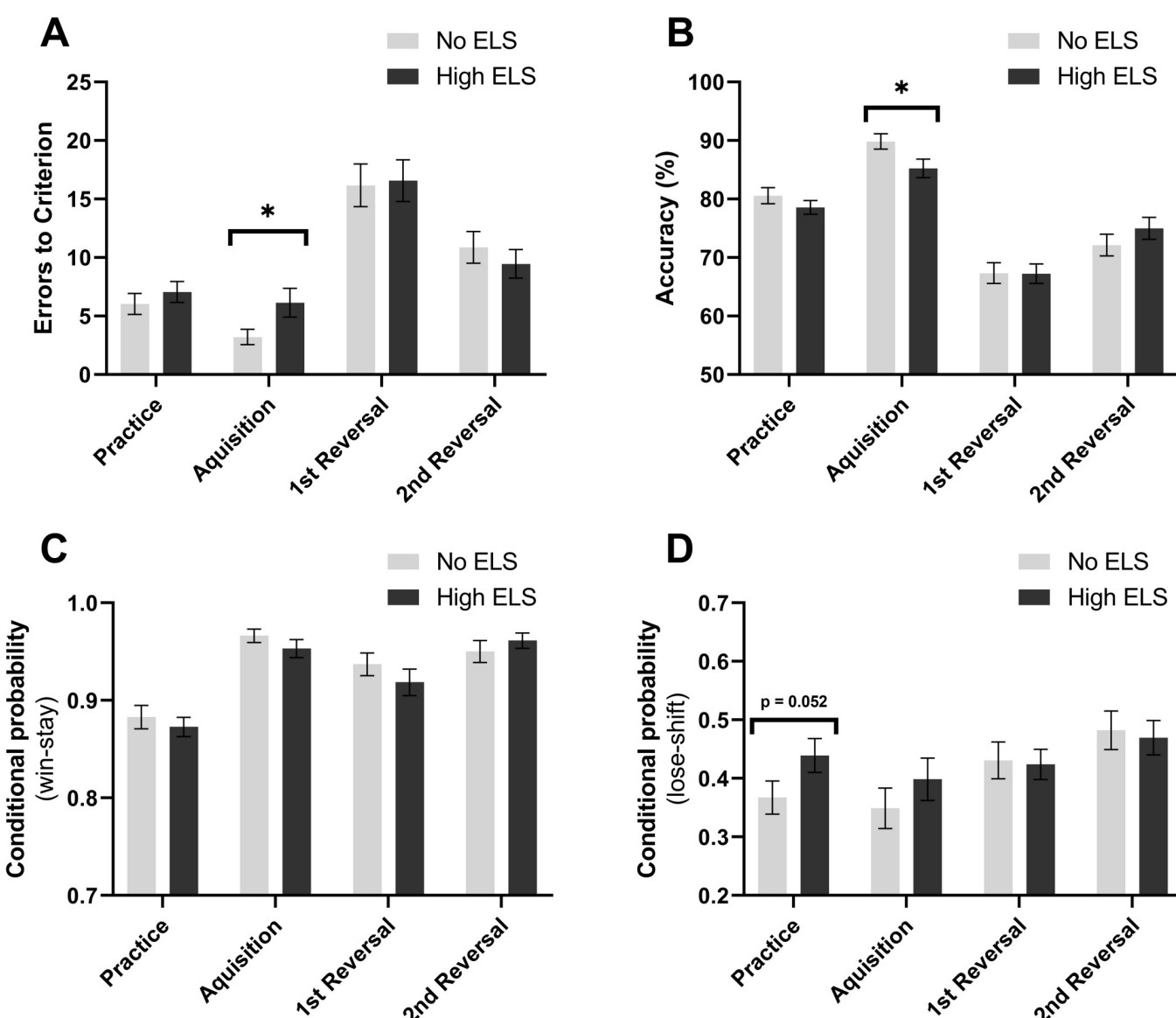

**Fig 3. High ELS participants show impaired learning in the acquisition phase of block 1.** **(A)** Errors made while reaching criterion for each phase, **(B)** accuracy within each phase, **(C** and **D)** win-stay and lose-shift probabilities for each phase of block 1 and practice respectively.

p = 0.19). Exploratory analysis further revealed a main effect of lifetime stress with higher lifetime stress corresponding to increased discrimination ability (GLMM, Z = 2.6, p = 0.007). An effect of gender was also revealed (GLMM, Z = 2.04, p = 0.04) with males showing increased discrimination ability. Finally, there was no difference between groups in response latencies (Fig 4D) nor was there an effect of stimulus upon response latency (Fig 4E).

Consistent with Pizzagalli and colleagues [21] the probability of misclassifying a stimulus based upon the preceding trial outcome was also analysed (S3 Table). Participants with a history of high levels of ELS were more likely to misclassify rich stimuli if either the previous trial was a not rewarded rich trial or a lean not rewarded trial with these measures roughly corresponding with rich lose-shift and lean lose-stay probability in the PRLT respectively.

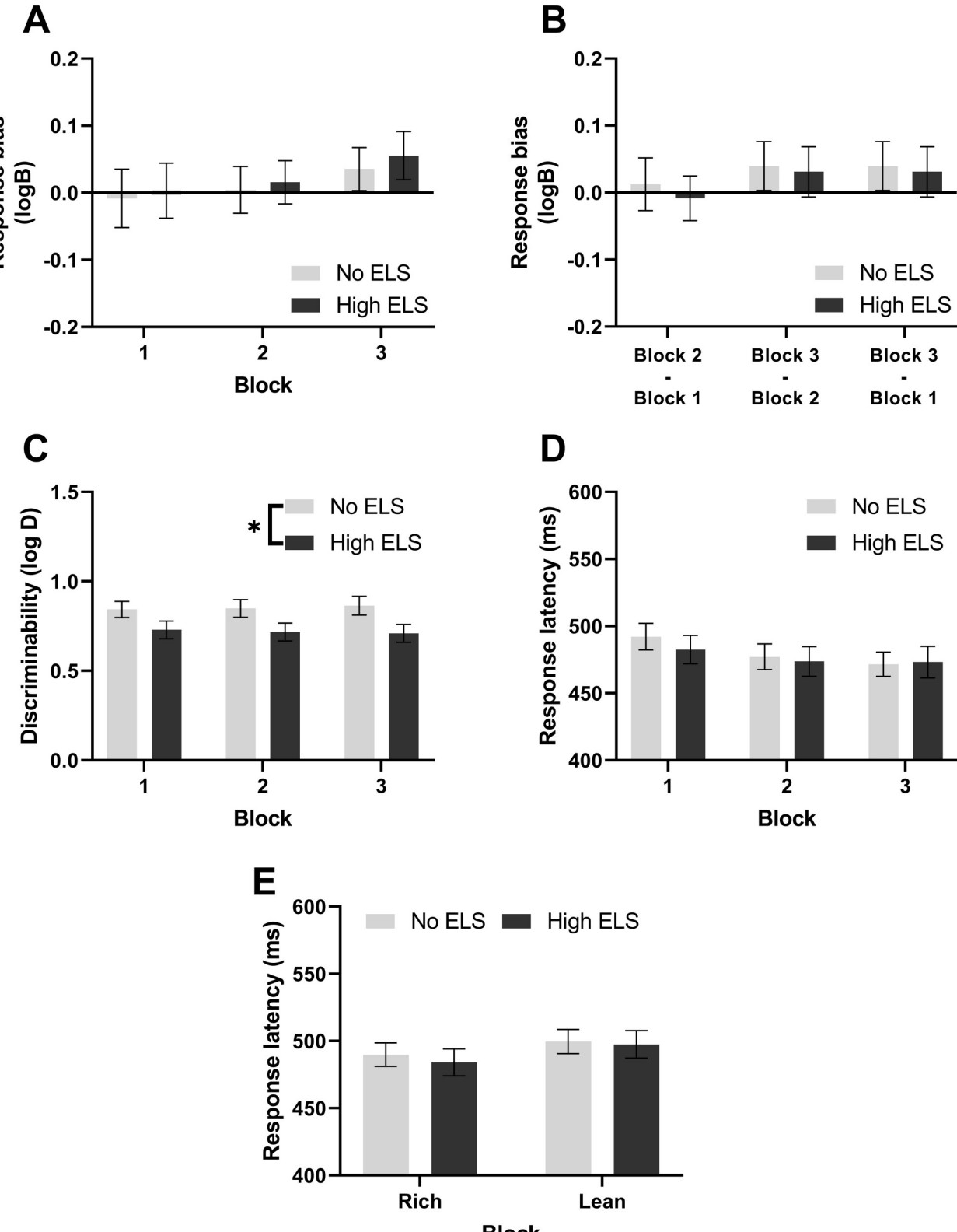

**Fig 4. Participants with a history of ELS show decreased discriminability in the PRT.** (A) Response bias to the more highly rewarded stimulus, (B) response bias development between blocks, (C) discriminability between long and short face lengths, (D) average response latency split by block and (E) response latency split by stimulus type.

## 4. Discussion

This study was designed to investigate whether healthy adults without a current psychiatric illness but with a history of ELS show alterations in reward processing and feedback sensitivity. To this end nearly 600 participants were screened with ELS being highly prevalent in the population with 79.0% of participants experiencing one or more ACE and 44.4% experiencing three or more.

Participants with a history of high ELS had higher self-report depression and anhedonia symptoms. Although participants stated they did not have a diagnosis of depression, 54.7% of high ELS and 26.2% of no ELS participants showed at least mild symptoms based upon the BDI-II questionnaire. BDI-II scores in no ELS (mean: 9.4 ± 1.0) and high ELS (mean: 15.2 ± 1.2) participants were higher than controls for similar studies [21,35,57] (range 1.3–3.62) but lower than depressed patients [21] (mean: 32.1 ± 8.6) or participants described as exhibiting a high BDI [57] (>16). These data are consistent with a large societal burden of undiagnosed depression [58–60]. It should be noted that the present study was undertaken during the Covid-19 global pandemic with it being estimated that levels of depression had doubled during this period [61]. The high number of participants reporting mild to severe symptoms of depression is a major limitation of this study as depression and anhedonia are well known to reduce reward learning in both the PRLT [29] and PRT [20,21,24]. It is also worth considering that 75% of adults with mental health conditions experience the onset of symptoms before aged 24 [62]. This means that the study population, all 25 years of age or greater, is potentially biased towards those more protected from mental health disorders. In keeping with large bodies of previous literature this study used retrospective reporting however it is worth noting the difficulties in correlating retrospective and prospective measures of ELS [63] suggesting that these might encompass different populations with different mechanistic links between ELS and depression vulnerability. Studies have also provided interesting insights when ELS is split into different modalities [64,65], however the present study was not powered enough to merit investigating this.

### 4.1 Probabilistic reversal learning task

In the PRLT participants with high ELS displayed decreased positive feedback sensitivity compared to controls as measured by win-stay probability. This finding was independent of depression symptomology and specific to PFS with no changes observed in lose-shift probability. This contrasts to depressed patients performing the PRLT who have been reported to show no change in positive feedback sensitivity [28,29]. Blunted striatal responses to reward in participants with a history of ELS have been previously reported [66,67] which speculatively may underlie the decreased PFS observed in the present study. Consistent with the present study, women with MDD and a history of childhood sexual abuse have also been found to have impaired performance in the PSST but only for trials requiring use of previously rewarded information and not those requiring use of previously punished information [35]. Within the PRLT depressed patients have been observed to show increased sensitivity to misleading negative feedback [28,29]. This was not observed in the high ELS cohort in the present study. In other tasks depressed patients have also been reported to show increased NFS alongside attenuated PFS [25,68–72]. These findings suggest that ELS is associated with changes in feedback sensitivity in the PRLT differently to depression with ELS decreasing PFS but not effecting NFS while depression has an opposite effect. It should however be noted that participants with ELS were more likely to have lifetime substance abuse [73] which may confound the results. Substance abuse is known to induce changes in feedback sensitivity and reward learning in addicts [74].

The PRLT also allows for assessment of reinforcement learning through the analysis of rule changes and accuracy in addition to parameters calculated through use of the reinforcement learning model. In contrast with our hypothesis, there was no evidence that ELS affected rule changes which is surprising considering evidence that both depression and ELS can impair cognitive flexibility [41,75]. The task in this study utilised performance-based reversals compared to previous ones using a fixed reversal interval therefore comparisons with depressed patients are challenging. However Murphy and colleagues did report that depressed patients made more errors following reversal [29]. Although rule changes were used as the main behavioural reward learning output, when data were analysed with the Q-learning model a trend towards decreased learning rate was observed in high ELS participants. This became significant in exploratory analysis and decreased associative learning has been previously observed in juveniles previously exposed to physical abuse [34]. There is a lack of consistent evidence in depression studies as to whether model free learning rate differs between patients and controls [76,77]. However, a recent study in a PRLT did report that depressed patients did show a decreased learning rate with this decreased learning rate being a better predictor of depression scores than simple task outputs. [72]. These findings warrant future investigation due to this study being only powered to detect group differences between two groups meaning that ANCOVA and exploratory analysis is likely to be underpowered.

A slower response latency was also observed in high ELS participants which was specific to the PRLT with no congruent changes seen in the PRT. This discrepancy may be related to differing cognitive demands with the PRLT potentially requiring greater working memory. Changes in response latency have not previously been observed in depressed patients carrying out the PRLT [28,29].

No directly comparable studies have been carried out in humans. However, maternally separated marmosets, an animal model of ELS, showed no change in simple visual discrimination compared to controls but showed impairments when the contingencies reversed [78]. This is similar to that seen in both depressed and bipolar patients in the human PRLT [29,79] who acquire the initial rule successfully but then are impaired following reversal. This compares to ELS participants in the present study who performed equally well in the practice phase and reversal phases but showed a deficit in acquisition of the first rule in block 1. This suggests a potential impairment in the ability to generalise the task rules between the practice and acquisition phase. However previous probabilistic learning studies did not include a practice phase meaning that this likely changed the way participants processed the start of block 1. This might explain the contrast with Pechtel and Pizzagalli, 2013 who reported that women with remitted MDD and ELS learnt acquisition in the PSST at the same rate as controls [35].

Exploratory analysis aimed to investigate if stress in adult life would modulate the relationship between reward processing deficits and ELS. There was little evidence for this except for an observed interaction between PFS and ELS whereby stress only influenced PFS in participants without a history of ELS. Higher lifetime stress led to greater PFS but higher stress in the last year was associated with decreased win-stay probability. There are few previous studies investigating similar constructs but Berghorst and colleagues reported that after stress induction those who had higher cortisol reactivity and self-reported negative affect had lower reward but not punishment sensitivity [33]. Additionally, it is worth noting that due to the relatively poor model fit for this exploratory analysis that these findings should be taken as preliminary due to the risk of data overfitting.

## 4.2 Probabilistic reward task

In contrast to previous studies employing the PRT neither ELS nor control groups showed a response bias toward the more highly rewarded stimulus [21,24] suggesting a general failure of

all participants to modulate their responses as a function of reward. This lack of response bias in the main study potentially indicates that the reward information was not salient enough therefore participants focussed upon correctly discriminating between mouth lengths. This makes comparison to previous literature challenging however it is worth noting that depressed patients have been reported to have a decreased response bias with no change in discrimination ability [20,21]. There are no previously published studies carrying out the PRT online. The online testing could have been one reason for the lack of response bias and likely leads to the high variability seen in the data as it is not possible to ensure that participants are completing the task in as controlled an environment as would be possible by laboratory testing.

However, another possibility for the lack of response bias seen was a key difference between laboratory and online versions of the task. While all other aspects of the task were similar, participants in the online task were informed that high performance would lead to a bonus payment with the actual reward in the task being points. Previous studies instead used direct monetary compensation in the task [24]. When the second population of control participants was tested in the PRT using direct monetary compensation a response bias was seen in blocks 1 and 3, however there did not appear to be evidence for this bias strengthening over time like previously observed [24]. There was also robust evidence for participants responding more quickly to the rich than lean stimulus as also has been reported [24,27]. While this difference in compensatory mechanism may underly the difference in control population performance in the two implementations of the task it could also be because the direct reward population had lower BDI and SHAPS scores. However, when the direct reward experiment was re-analysed with no BDI and SHAPS cut-offs such that it more closely approximated the control population in the main study the results were much the same as with the cut-offs (S5 Fig). These data therefore suggest that it is possible to successfully implement the PRT in an online setting using the directly rewarded task with the availability of reliable online psychological tasks being key under current circumstances.

While difficult to interpret for previously discussed reasons, participants with high levels of ELS did show impairments in discrimination, a measure of task difficulty. This however appeared to be driven by changes in depression symptomology as opposed to ELS specifically.

## 4.3 Conclusions

These data suggest that participants who do not self-report a diagnosis of a mental health condition but do have a history of ELS show impairments in positive feedback sensitivity and reward learning in the PRLT compared to controls. These impairments may be important in understanding how ELS predisposes to depression with reduced reward learning being a key feature in MDD patients [22]. However, high levels of potentially undiagnosed depression are a potential confound and highlight a potential wider issue in terms of the number of people who meet criteria for MDD but are not formally diagnosed or receiving care. Future studies are needed to replicate these findings, investigate the neural circuit changes underlying these reward learning impairments and investigate whether these findings are directly related to psychiatric risk.

## Supporting information

**S1 Fig. Study overview.** Participants were screened by ELSQ score and then formed into two study groups: no ELS and high ELS.
(PNG)

**S2 Fig. Early life stress in an online study population.** (A) ELSQ scores in the study population. (B) Mental health disorder/Parkinson's self-report diagnosis by ELSQ score (Mann-Whitney, U = 15725, p < 0.0001). N = 586 participants. (C) ELSQ scores in the study population split by modality of adverse childhood experience.
(TIF)

**S3 Fig. Interpretation of BDI-II and SHAPS scores in the no and high ELS populations.** Scores were interpreted following Beck et al., 1996 and Snaith et al., 1995. (A) BDI-II split by severity of depression (chi2, $\chi2(3) = 12.9$, p = 0.005) and (B) SHAPS split by normal or abnormal hedonic responses (chi2, $\chi2(1) = 6.3$, p = 0.012). N = 129 participants (65 no ELS, 64 high ELS).
(TIF)

**S4 Fig. Direct monetary reward in the PRT using a control population led to a reward induced bias.** (A) While no overall effect of block was observed, a response bias was observed in blocks 1 and 3 (Wilcoxon signed ranks test, block 1: W = 1087.5, p = 0.001, block 3: W = 916.5, p = 0.038). (B) There was little evidence for response bias strengthening across blocks. Discriminability (C) and response latency (D) did not appear to change over the course of a session. (E) Participants were faster to respond to the rich stimulus than lean (Wilcoxon matched pairs signed ranks test, W = 814, p = 0.0007). N = 56 participants.
(TIF)

**S5 Fig. Direct monetary reward in the PRT using a control population led to a reward induced bias (no participants excluded due to BDI or SHAPS scores).** (A) While no overall effect of block was observed, a response bias was observed in blocks 1 and 3 (Wilcoxon signed ranks test, block 1: W = 1174, p = 0.003, block 3: W = 1105, p = 0.004). (B) Response bias strengthened between blocks 3 and 2 (Wilcoxon signed ranks test, W = 873, p = 0.040). Discriminability (C) and response latency (D) did not appear to change over the course of a session. (E) Participants were faster to respond to the rich stimulus than lean (Wilcoxon matched pairs signed ranks test, W = 1372, p < 0.0001). N = 81 participants.
(TIF)

**S1 Table. Principal component analysis of social scale, SHAPS and BDI-II scores.** The mean ± standard error are shown for each group with the relevant statistical comparison.
(DOCX)

**S2 Table. Principal component analysis component loadings.**
(DOCX)

**S3 Table. Miss-rates, the chance of mis-categorising a stimulus, by previous trial.** Data is shown as mean ± standard error and significant p-values are shown in bold.
(DOCX)

**S4 Table. Demographic and self-report measures in the directly rewarded PRT control population.** Values are shown for each group as mean ± standard error where appropriate.
(DOCX)

**S5 Table. Comparison of ANOVA/ANCOVA analytical approach used as main analysis with results obtained using generalised linear mixed models.** Measures by task phase were analysed as multiple Mann-Whitney tests due to non-parametricity of data. #1: Acquisition, Z = -1.38, p = 0.17, First reversal, Z = -2.0, p = 0.045, #2: Acquisition, Z = -1.44, p = 0.15, First reversal, Z = -2.1, p = 0.036, #3: Acquisition, Z = -1.41, p = 0.16, First reversal, Z = -1.2,

p = 0.24, #4: Acquisition, Z = -1.95, p = 0.05, First reversal, Z = -0.92, p = 0.36.
(DOCX)

**S1 File. Supplementary methods for the directly rewarded PRT in a control population.**
(DOCX)

## Acknowledgments

The authors would like to thank Michelle Taylor for advice on statistical analysis.

## Author Contributions

**Conceptualization:** Matthew Paul Wilkinson, Jack Robert Mellor, Emma Susan Jane Robinson.

**Data curation:** Matthew Paul Wilkinson, Chloe Louise Slaney.

**Formal analysis:** Matthew Paul Wilkinson, Chloe Louise Slaney.

**Funding acquisition:** Jack Robert Mellor, Emma Susan Jane Robinson.

**Investigation:** Matthew Paul Wilkinson, Chloe Louise Slaney.

**Methodology:** Matthew Paul Wilkinson, Emma Susan Jane Robinson.

**Project administration:** Matthew Paul Wilkinson.

**Resources:** Jack Robert Mellor, Emma Susan Jane Robinson.

**Software:** Matthew Paul Wilkinson.

**Supervision:** Jack Robert Mellor, Emma Susan Jane Robinson.

**Validation:** Matthew Paul Wilkinson.

**Visualization:** Matthew Paul Wilkinson.

**Writing – original draft:** Matthew Paul Wilkinson.

**Writing – review & editing:** Matthew Paul Wilkinson, Chloe Louise Slaney, Jack Robert Mellor, Emma Susan Jane Robinson.

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
