## [Decision Letter · Decision Letter 0]

23 Jul 2021

PONE-D-21-19019

Investigation of reward learning and feedback sensitivity in non-clinical participants with a history of early life stress.

PLOS ONE

Dear Dr. Wilkinson,

Thank you for submitting your manuscript to PLOS ONE. After careful consideration, we feel that it has merit but does not fully meet PLOS ONE’s publication criteria as it currently stands. Therefore, we invite you to submit a revised version of the manuscript that addresses the points raised during the review process.

We look forward to receiving your revised manuscript.

Kind regards,

Alexandra Kavushansky, PhD

Academic Editor

PLOS ONE

Journal Requirements:

“This work was primarily funded by the BBSRC SWBio DTP PhD programme (grant numbers: BB/J014400/1 and BB/M009122/1) awarded to MPW. Additional support was also provided from the Wellcome Trust Neural Dynamics PhD studentship (grant number: 108899/B/15/Z) awarded to CLS”

 “This work was primarily funded by the BBSRC SWBio DTP PhD programme (www.bbsrc.ukri.org, grant numbers: BB/J014400/1 and BB/M009122/1) awarded to MPW. Additional support was also provided from the Wellcome Trust Neural Dynamics PhD studentship (www.wellcome.org, grant number: 108899/B/15/Z) awarded to CLS. The funders had no role in study design, data collection and analysis, decision to publish, or preparation of the manuscript.”

Additional Editor Comments:

In addition to the issues raised by the reviewers, I'll like to ask to add the information about the test that was used to verify normal distribution of the data.

Reviewers' comments:

Reviewer's Responses to Questions

**Comments to the Author**

1. Is the manuscript technically sound, and do the data support the conclusions?

Reviewer #1: Yes

Reviewer #2: Partly

2. Has the statistical analysis been performed appropriately and rigorously? 

Reviewer #1: Yes

Reviewer #2: No

3. Have the authors made all data underlying the findings in their manuscript fully available?

Reviewer #1: Yes

Reviewer #2: Yes

4. Is the manuscript presented in an intelligible fashion and written in standard English?

Reviewer #1: Yes

Reviewer #2: No

5. Review Comments to the Author

Reviewer #1: This manuscript describes an online study using two reinforcement learning tasks, comparing adult individuals reporting high levels of early-life adversity (ELA) and those who do not. While ELA is a risk factor in the development of depression, the current study excluded those reporting a psychiatric diagnosis. A motivation for the study was to investigate whether ELA may lead to a deficit reward processing that could be seen as a vulnerability to developing depression. A strength of the current study was the measurement of several demographic factors related to depression, stress, and socio-economic status that enabled the authors to use regression to examine the effects of ELA independently. One weakness that should be mentioned as a caveat is the fact that lifetime substance use is likely to have varied between the groups and could have contributed to the results.

The main finding was the demonstration of a reduced reward sensitivity (win-stay) behavior in participants in the probabilistic reversal learning task. This effect remained even when accounting for current levels of depression. There was no effect on reversal learning, and thus cognitive flexibility did not seem affected. There was a trend for a slower learning rate, but this was fairly subtle. Overall, the selectivity of the deficit in reward processing was thus interesting.

It appears that the implementation of the second task (probabilistic reward) did not yield interpretable results in the online format. The second control group showed that using actual monetary reward improved reward sensitivity in this online format, and thus it would be more informative to use this version in a future study examining ELA. As it stands, I feel that the data from this task do not add much as neither group showed reward sensitivity in this implementation.

Reviewer #2: The paper “Investigation of reward learning and feedback sensitivity in non-clinical participants with a history of early life stress” examines the relationships between early stressful event exposures and multiple facets of reward learning. While an important question with potentially interesting findings, the manuscript in its current form lacks the theoretical and methodological clarity necessary to interpret the presented results.

1. On a whole, the manuscript is difficult to read and follow. I found myself having to search for clarifying details, finding them in places you would not typically expect – i.e. methods in the introduction or discussion – or not all. The paper in its current form requires extensive re-writing and clarification before I can effectively evaluate the presented results. I have provided more concrete suggestions for this could be addressed below.

2. The introduction lacks strong theoretical justification for study. This is not because there is no reason to believe the question would be one of interest, but rather due to the current structure of the introduction.

a. The introduction sets the paper up as though it will be examining whether alterations in reward learning processes moderate the relationship between early life stress (ELS) and depression. However, this is not what is tested in the results – the primary analyses of interest focus on the relationship between ELS and learning parameters, while controlling for depressive symptoms. The introduction would benefit from a more extensive review of the existing evidence related to ELS and reward learning – an area that has grown rapidly over the last 10 years (for review see Herzberg & Gunnar, 2020) – and a more thorough explanation of how the current manuscript addresses an outstanding question within that literature.

Herzberg & Gunnar (2020). Early life stress and brain function: Activity and Connectivity associated with processing emotion and reward. Neuroimage, 209: 116493.

b. The introduction uses a lot of jargon without clarifying what these terms refer to and why they may be of interest. For example, authors discuss the effects of depression on “the probabilistic reward learning” and “the probabilistic reversal learning task” on page 3, but provide little information on what the specific tasks referred are testing or why the findings are of interest. It’s important to provide this information to help justify the current study, especially as many variations of probabilistic reward learning tasks have been used in the literature examining early life stress and reward learning.

c. The authors should provide more specific hypotheses than just ELS will be associated with altered reward processing and feedback sensitivity. There is enough existing literature to allow for concrete predictions. The justification for each hypothesized effect should also be made concrete.

d. Relatedly, despite the fact the authors utilize a probabilistic reversal learning task, they include no discussion of existing research examining reversal learning/cognitive flexibility and ELS. Research finding ELS is associated with less cognitive flexibility during reversal learning (see Harms et al, 2017) and different learning strategies (see Letkiewicz et al, 2020) is directly relevant to the questions being asked here.

Harms et al (2017). Instrumental learning and cognitive flexibility processes are impaired in children exposed to early life stress. Developmental Science, 21(4): e12596.

Letkiwicz et al (in press). Frontoparietal network activity during model-based reinforcement learning updates is reduced among adolescents with sever sexual abuse. Journal of Psychiatric Science.

3. The Methods are also unclear and difficult to follow making it hard to interpret the presented results.

a. It is not clear why the authors used repeated measures ANCOVA for some of the analyses and GLMMs for other. GLMMs have no issue handling repeated measures data. The authors should justify why they used the different approaches for different outcomes.

b. It is unclear what the follow up study is addressing. In the introduction, it is stated that it was “due to the failure of both groups to meet the primary endpoint” and then in the methods and discussion it is stated this is because neither group showed a response bias towards the high reward option. Why the follow up study is helpful needs to be made clearer and more explicit. This information should reside in the Methods, not interspersed throughout the Introduction, Methods, and Discussion. Additionally, I would recommend moving the follow up study in its entirety to the Supplemental Materials as it doesn’t appear to be directly relevant to the primary question of interest.

c. It would be helpful to provide more information about what the model parameters from the Q learning model represent. Readers familiar with computational learning models will know this, but more naieve readers may not. This can be as simple as explaining the learning rate represents speed of feedback integration and will also help readers better interpret the results.

4. The results should be re-written in line with the introduction to make it clear how the different analyses speak to the hypotheses of interest. As currently written, the findings seem somewhat spurious and it is unclear how meaningful they are.

More minor comments

1. Are the effects similar if you run the analyses treating ELS as continuous rather than dichotomous?

2. The manuscript should be thoroughly read for typos and grammatical errors.

6. PLOS authors have the option to publish the peer review history of their article (what does this mean?). If published, this will include your full peer review and any attached files.

Reviewer #1: **Yes: **Barbara Knowlton

Reviewer #2: No

---

## [Author Response · Author response to Decision Letter 0]

3 Sep 2021

We appreciate the time taken in reviewing this manuscript. Please find a detailed response to every comment raised in the included cover letter.

---

## [Decision Letter · Decision Letter 1]

20 Sep 2021

PONE-D-21-19019R1Investigation of reward learning and feedback sensitivity in non-clinical participants with a history of early life stress.PLOS ONE

Dear Dr. Wilkinson,

Thank you for submitting your manuscript to PLOS ONE. After careful consideration, we feel that it has merit but does not fully meet PLOS ONE’s publication criteria as it currently stands. Therefore, we invite you to submit a revised version of the manuscript that addresses the points raised during the review process.

We look forward to receiving your revised manuscript.

Kind regards,

Sarah Whittle

Academic Editor

PLOS ONE

Journal Requirements:

Reviewers' comments:

Reviewer's Responses to Questions

**Comments to the Author**

1. If the authors have adequately addressed your comments raised in a previous round of review and you feel that this manuscript is now acceptable for publication, you may indicate that here to bypass the “Comments to the Author” section, enter your conflict of interest statement in the “Confidential to Editor” section, and submit your "Accept" recommendation.

Reviewer #2: (No Response)

2. Is the manuscript technically sound, and do the data support the conclusions?

Reviewer #2: Partly

3. Has the statistical analysis been performed appropriately and rigorously? 

Reviewer #2: No

4. Have the authors made all data underlying the findings in their manuscript fully available?

Reviewer #2: Yes

5. Is the manuscript presented in an intelligible fashion and written in standard English?

Reviewer #2: No

6. Review Comments to the Author

Reviewer #2: Thank you to the authors for their detailed responses and revisions. The revisions have improved the manuscript, particularly in regards to clarifying the motivation for the study and the methods. However, there are still a few issues that could be addressed to improve the work.

1. I appreciate the authors’ rationale for their use of the different analytical techniques for the different analyses (ANOVA, ANCOVA, GLMM). However, I still have concerns about the use of the different techniques – in particular, it makes it difficult to determine whether differences in the effects of ELS in the ANOVA/ANCOVA and GLMM analyses are due to the incorporation of additional predictors in the GLMM or the different analytical techniques. Could the authors re-run all analyses using GLMMs to help clarify this? As long as the results are comparable, I do not think these need to be reported in the main manuscript, but they would be helpful as Supplemental for readers with similar concerns.

2. While the revisions have improved the readability of the manuscript, there are still areas where the language is unclear making it difficult to follow. I advise the authors read through the manuscript carefully to improve readability and flow of the text. Some specific examples of where this would be useful are outlined below:

a. The abstract could be revised for clarity. In particular, the second sentence with the hypothesis is confusing unless you’ve read the paper already – the way it is written makes it sound like the hypothesis is contradictory to current literature (i.e. it’s stated that depression is associated with reward deficits but then the hypothesis is that ELS will be associated with reward deficits but depression will not). Additionally, given the study the modified PRT in control participants has been moved to the supplemental, it’s not clear this is necessary in the abstract.

b. I recommend explicitly stating the goal of the study before introducing the hypothesis in the last paragraph of the introduction. The goal of assessing the relationship between ELS and reward processing using tasks that can be directly compared with those that have been used in the literature on depression is an important one and it will help readers to have that made clear before reading the methods and results. Additionally, it might help to outline the three sub questions (Does ELS affect reward learning parameters, is this due to depression, and does ELS interact with acute stress) here to orient the readers to how the results will be conveyed.

c. Could you speak to how the results tie to the work that has been done using these tasks in patients with depression in the discussion as that is one of the motivations for the study outlined in the introduction?

7. PLOS authors have the option to publish the peer review history of their article (what does this mean?). If published, this will include your full peer review and any attached files.

Reviewer #2: No

---

## [Author Response · Author response to Decision Letter 1]

4 Nov 2021

Please see attached cover letter providing detailed responses to each comment by the reviewers.

---

## [Editor Report · Decision Letter 2]

10 Nov 2021

Investigation of reward learning and feedback sensitivity in non-clinical participants with a history of early life stress.

PONE-D-21-19019R2

Dear Dr. Wilkinson,

We’re pleased to inform you that your manuscript has been judged scientifically suitable for publication and will be formally accepted for publication once it meets all outstanding technical requirements.

Kind regards,

Sarah Whittle

Academic Editor

PLOS ONE
---

## [Editor Report · Acceptance letter]

3 Dec 2021

PONE-D-21-19019R2 

Investigation of reward learning and feedback sensitivity in non-clinical participants with a history of early life stress. 

Dear Dr. Wilkinson:

I'm pleased to inform you that your manuscript has been deemed suitable for publication in PLOS ONE. Congratulations! Your manuscript is now with our production department. 

Kind regards, 

on behalf of

Dr. Sarah Whittle 

Academic Editor

PLOS ONE